# Non-resonant light scattering in dispersions of 2D nanosheets

Andrew Harvey[1,2], Claudia Backes[3], John B. Boland[1,2], Xiaoyun He[1,2], Aideen Griffin[1,2], Beata Szydlowska[1,2], Cian Gabbett[1,2], John F. Donegan[1,2] & Jonathan N. Coleman[1,2]

Extinction spectra of nanomaterial suspensions can be dominated by light scattering, hampering quantitative spectral analysis. No simple models exist for the wavelength-dependence of the scattering coefficients in suspensions of arbitrary-sized, high-aspect-ratio nanoparticles. Here, suspensions of BN, talc, GaS, $Ni(OH)_2$, $Mg(OH)_2$ and $Cu(OH)_2$ nanosheets are used to explore non-resonant scattering in wide-bandgap 2D nanomaterials. Using an integrating sphere, scattering coefficient ($\sigma$) spectra were measured for a number of size-selected fractions for each nanosheet type. Generally, $\sigma$ scales as a power-law with wavelength in the non-resonant regime: $\sigma(\lambda) \propto [\lambda/\langle L \rangle]^{-m}$, where $\langle L \rangle$ is the mean nanosheet length. For all materials, the scattering exponent, $m$, forms a master-curve, transitioning from $m = 4$ to $m = 2$, as the characteristic nanosheet area increases, indicating a transition from Rayleigh to van der Hulst scattering. In addition, once material density and refractive index are factored out, the proportionality constant relating $\sigma$ to $[\lambda/\langle L \rangle]^{-m}$, also forms a master-curve when plotted versus $\langle L \rangle$.

[1] CRANN & AMBER Research Centers, Trinity College Dublin, Dublin 2, Ireland. [2] School of Physics, Trinity College Dublin, Dublin 2, Ireland. [3] Chair of Applied Physical Chemistry, University of Heidelberg, Im Neuenheimer Feld 253, 69120 Heidelberg, Germany. Correspondence and requests for materials should be addressed to J.N.C. (email: colemaj@tcd.ie)

In addition to being an important area of basic physics, the interaction of light with matter is the basis for optical characterisation of materials. Various types of spectroscopy are used in chemistry, materials science and nanoscience with special emphasis on optical characterisation of solutions or suspensions. Absorption spectra are particularly useful, yielding information about resonant optical transitions, allowing concentration measurements, identification of individual species in mixed suspensions[1] and even providing information on particle dimensions[2,3]. However, measuring absorption spectra can pose unexpected challenges. Absorption is usually measured via optical extinction spectroscopy where the extinction, *Ext*, is defined via the transmittance: $T = 10^{-Ext}$. In fact, extinction spectra always contain contributions from both absorption and scattering. Only for molecular solutions and suspensions of small nanoparticles, is the scattering weak enough to be ignored, allowing researchers to equate the absorption and extinction spectra. Conversely, for high-aspect-ratio nanomaterials (with lateral sizes of >50 nm), extinction spectra can be dominated by wavelength-dependent scattering contributions[2,4] which complicate quantitative analysis and are relatively poorly understood.

Scattering contributions manifest themselves as a broad background which falls off with increasing wavelength in the non-resonant regime[2,5]. Experimentally, the scattering intensity increases with increasing particle size[2,5], with some suspended 2D nanosheets displaying scattering backgrounds which are considerably more intense than the absorption[5,6]. Then, the presence of this scattering background can obscure or shift the positions of absorption peaks, making detailed analysis difficult[5,6]. Because the scattering intensity varies with particle size in a poorly understood way, extracting solute concentration from extinction spectra is challenging[5]. While light-scattering effects can be removed by using an index-matched solvent in certain cases, this is often impossible for solvent-stabilised suspensions of nanosheets, the stability of which is sensitive to solvent type[7,8].

While some aspects of light scattering (LS) from nanoparticles, such as its angular dependence, are very well understood[9], much less attention has been paid to its spectral dependence[10–13], especially for highly anisotropic scattering particles such as nano-rods or nano-platelets. Thus, lack of knowledge about the scattering component makes it difficult to analyse the extinction spectra of suspensions of nano-platelets such as 2D nanosheets.

LS is a ubiquitous phenomenon where light rays are deviated from their incident trajectory through interactions with local non-uniformities in refractive index[14]. LS is usually elastic with the main effect being the reduction in intensity of a light beam in the forward direction as light is scattered out of the beam. The most well-known example of scattering is Rayleigh scattering, which is appropriate for very small (size $< \lambda/10$) scattering particles (e.g. molecules).

While LS is usually measured as a function of scattering angle[9], we are interested in its wavelength-dependence due to its contribution to measured extinction spectra. Using standard optical spectrometers in transmission mode, the measured extinction (*Ext*) always has contributions from both absorbance (*Abs*) and scattering (*Sca*)[14]:

$$\log T = -Ext = -(Abs + Sca) \tag{1}$$

Converting to optical coefficients (e.g. $Ext/l = \varepsilon C$ where $l$ is the cell length, $\varepsilon$ is the extinction coefficient and $C$ is the concentration of dispersed objects [g/L]) yields[14]:

$$\varepsilon(\lambda) = \alpha(\lambda) + \sigma(\lambda) \tag{2}$$

where $\alpha(\lambda)$ and $\sigma(\lambda)$ are the absorption and scattering coefficients, respectively (units L/g/m). In molecular solutions, scattering can

generally be neglected. To illustrate this, we can use Rayleigh scattering to estimate the scattering coefficient for molecular scatterers. Modelling the scatterers as spheres of diameter $D$, the Rayleigh scattering coefficient can be found to be[14]:

$$\sigma(\lambda) = \frac{4\pi^4}{\rho D}\left[\frac{(n/n_0)^2 - 1}{(n/n_0)^2 + 2}\right]^2\left(\frac{\lambda}{D}\right)^{-4} \tag{3}$$

where $\rho$ is the sphere density and $n$ and $n_0$ are the refractive indices inside and outside the sphere (the scattering coefficient is the scattering cross section, $\sigma_{CS}$, divided by the sphere mass: $\sigma = \sigma_{CS}/(\rho V)_{sphere}$). Taking $\rho = 1000$ kg/m³, $n = 1.5$, $n_0 = 1.3$ and $D = 0.5$ nm shows that in a molecular solution, $\sigma$ is very small, $<10^{-5}$ L/g/m over the entire visible spectrum. This is much smaller than values of $\alpha$ typically found for molecular systems[15], allowing $\sigma$ to be neglected in such systems and leading to the common misconception that extinction and absorption are interchangeable.

However, even though Eq. (3) only holds for $D < \lambda/10$, it clearly suggests that the magnitude of the scattering coefficient increases strongly with the size of the scattering objects, implying that dispersions of nanoparticles with $D \sim \lambda$ might display scattering which is non-negligible compared to absorption. Although this phenomenon is rarely referred to, one of the first demonstrations of liquid-exfoliated BN, $MoS_2$ and $WS_2$ nanosheets was accompanied by the observation that extinction spectra clearly contained a power-law scattering component[16]. Subsequently Yadgarov et al.[4] and Backes et al.[2] showed that, when absorption and scattering spectra are separated using an integrating sphere, dispersions of $MoS_2$ nano-platelets (size ~100 nm) display scattering coefficient spectra which are similar in magnitude to the absorption coefficient spectra in the resonant regime. This was followed by similar results for a small number of 2D materials, namely graphene, black phosphorous, SnO, $Ni(OH)_2$ and GaS[5,6,17–20]. Clearly, this means that scattering cannot be neglected in such systems. However, very little is known about the details of scattering coefficient spectra for nanoparticles with dimensions close to the wavelength of light.

In principle, the scattered intensity from any object can be calculated using Mie scattering theory[14]. The overall procedure is mathematically complex and is typically performed computationally for particles of arbitrary size and shape. For small ($D < \lambda/10$), spherical scattering objects, a number of approximations can be made which yield Eq. (3) (Rayleigh scattering)[14]. In addition, as discussed below, there is a very good approximation associated with larger spherical particles (see Eq. 5). However, there are no solutions to Mie theory which yield simple closed-form expressions for the scattering coefficient of dispersions of discs or rods with long dimension $>\lambda/10$. This makes it challenging to understand the nature of scattering coefficient spectra for nanosheets. In fact there are very few papers in the literature which measure and discuss scattering spectra for disk-like objects. This is a significant gap, especially given the current technological importance of 2D materials[21,22].

The aim of this paper is to measure non-resonant LS for a range of 2D nanomaterials so as to understand what parameters control the scattering coefficient spectra. To do this, an appropriate material set is required with each material displaying a wide bandgap to facilitate scattering measurements over a broad wavelength range. In addition, it should be possible to disperse each material in liquids[23], to obtain optical measurements on dispersed samples. The liquids themselves must have very low absorption in the spectral region of the measurements. We selected BN[24], talc[25], GaS[6] and three members of the 2D metal hydroxide family; $Ni(OH)_2$[5], $Mg(OH)_2$ and $Cu(OH)_2$. Each of

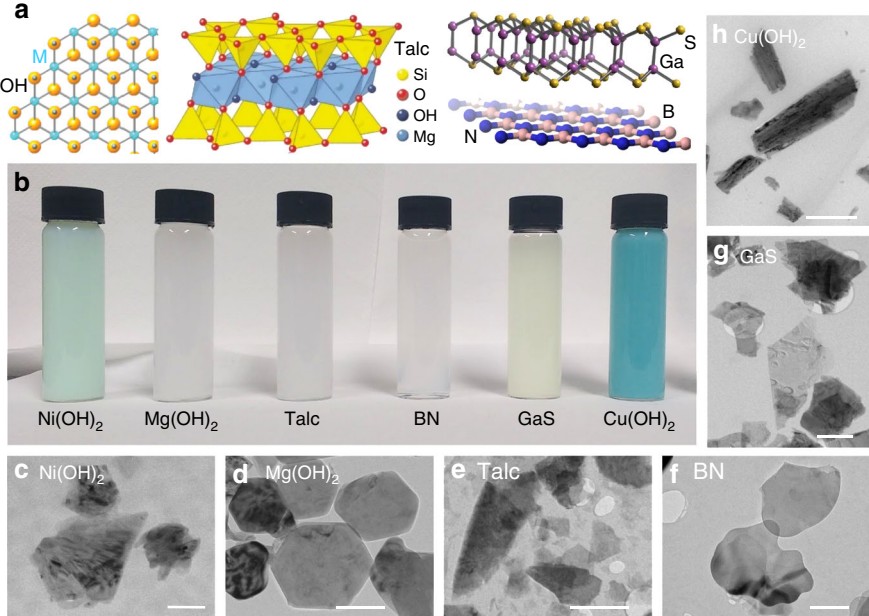

**Fig. 1** Exfoliation of insulating 2D materials. **a** Structure of materials used. Left-right: metal hydroxides (M(OH)$_2$ where M = Ni, Mg or Cu in this work), talc (Mg$_3$Si$_4$O$_{10}$(OH)$_2$), GaS and BN. **b** Photograph of dispersions of all six materials. In each case, these are very large (XL), size-selected nanosheets which show strong scattering. **c–h** Representative TEM images of few-layer nanosheets of each type (in each case extracted from XL fractions). In panels (**c**, **d**), the scale bar is 200 nm while in (**e–h**), it is 500 nm

these materials have bandgaps above ~3 eV, giving us a wide spectral range to probe non-resonant scattering. Of these, the first four have previously been produced by exfoliation of the parent crystal in liquids, as indicated by the references. Because of their similar surface chemistry, the last two materials should be straightforward to exfoliate using techniques applied to Ni(OH)$_2$. After applying liquid-exfoliation[23] to the six 2D materials, the exfoliated dispersions were then size-selected to yield fractions containing nanosheets of different sizes, as confirmed by TEM and AFM measurements. For each of the resultant nanosheet dispersions, we measured the extinction coefficient spectra in a standard optical spectrometer and then the absorption spectra using an integrating sphere. The scattering spectra could then be obtained by subtracting absorbance from extinction and were analysed within the framework of scattering theory, yielding a number of insights into LS by 2D materials.

## Results

### Basic exfoliation and characterisation of wide-bandgap 2D materials

The structures of the 2D materials studied in this work are shown in Fig. 1a. Talc, GaS and BN were exfoliated according to the published procedures[6,25,26]. For the previously unexfoliated materials, Mg(OH)$_2$ and Cu(OH)$_2$, we applied a standard liquid phase exfoliation protocol previously used to exfoliate Ni(OH)$_2$ nanosheets[5]. In all cases, the starting materials were purchased commercially as powders (see Methods). In brief, washed powders of all six starting materials were added to water and sodium cholate and sonicated in a metal beaker using an ultrasonic tip. The exception is GaS which was exfoliated in N-methyl-2-pyrrolidone (NMP) using a sonic bath due to its propensity to oxidise in water[6]. The dispersions were then size-selected, as described in more detail below, producing ~5 distinct fractions. In Fig. 1 the fraction with the largest nanosheets is shown, yielding dispersions with a range of colours from pale green to blue (Fig. 1b). We confirmed that all exfoliated nanosheets were of the expected material via Raman spectroscopy (Supplementary Fig. 1, Supplementary Note 1).

Transmission electron microscopy characterisation (Fig. 1c–h) showed all six dispersion types to contain large quantities of nanosheets with no non-2D material present. For all materials except Cu(OH)$_2$, the exfoliated product consisted of 2D nanosheets similar to those previously produced by LPE[18,27–29]. As expected, the as-prepared dispersions contained few-layer nanosheets with a broad thickness distribution[3,6,18,29,30]. While such thickness polydispersity is clearly a disadvantage over alternative methods of layered hydroxide exfoliation such as ion-exchange, we believe it to be an acceptable price to pay for the versatility, speed and ease associated with LPE. As shown in Fig. 1h, the Cu(OH)$_2$ sample was clearly different to the other materials, displaying a more belt-like structure. While the length/width aspect ratio for the other five materials was close to ~1.5, the Cu(OH)$_2$ nanosheets displayed length/width ~4 with the reasons for this difference currently unclear.

### Size selection of wide-bandgap 2D materials

Typically, as-prepared dispersions produced by LPE contain nanosheets with a wide range of lateral sizes and thicknesses. This inherent poly-dispersity allows the nanosheets to be size-selected into fractions using LCC[3], a method that involves a number of sequential centrifugation steps, each using an increased centrifugation speed, to isolate nanosheets in different size ranges (Methods). In brief, each fraction is produced by centrifuging a supernatant obtained from a previous step at a given relative centrifugal force (RCF, expressed in units of the earth's gravitational field g) to remove the largest nanosheets into the sediment. The sediment is then separated and retained while the supernatant is then centrifuged at a higher RCF, again resulting in the largest remaining nanosheets entering the sediment. Repeating multiple times yields a set of sediments containing nanosheets of different sizes. The sediments can easily be redispersed as desired. Here, we typically produced five or more size-selected dispersions for each of the six materials. The fractions can be quantitatively differentiated by the "central g-value", that is the midpoint of the two centrifugation accelerations used to produce each fraction. However, for

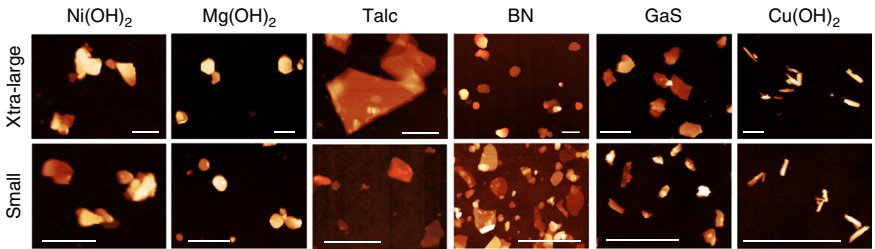

**Fig. 2** AFM characterisation of liquid-exfoliated nanosheets. Representative AFM images of size-selected 2D materials showing (top row) the extra-large (XL) fraction and (bottom row) the small (S) fraction. Sample images are shown in the Supplementary Figure 7 with z-axis scale bars. Spatial scale bars are as follows: $Ni(OH)_2$, 200 nm; Talc, 500 nm; $Mg(OH)_2$, BN, GaS, $Cu(OH)_2$, 1 μm

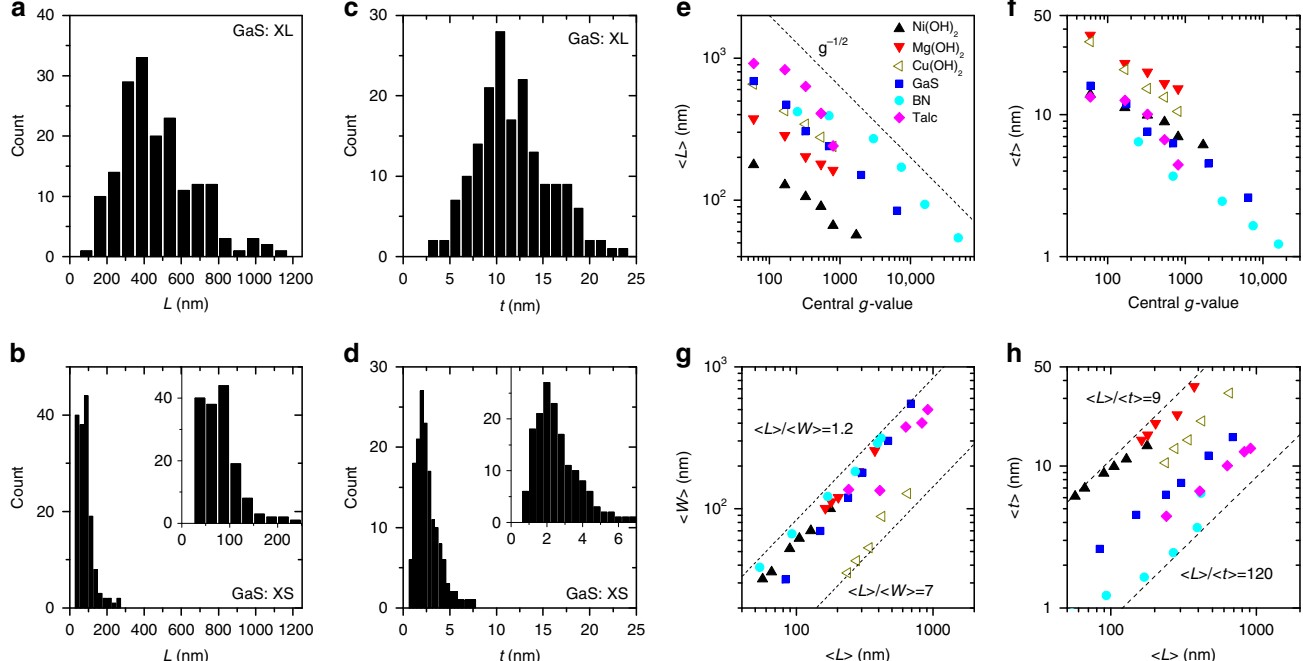

**Fig. 3** Dimensions of size-selected nanosheets as extracted from AFM. **a**, **b** Histograms showing nanosheet length (the longest dimension) from the XL and XS fractions. **c**, **d** Histograms showing nanosheet thickness from the XL and XS fractions. The insets in **b** and **d** show magnified versions of the XS data. **e**, **f** Mean nanosheet length (**e**) and thickness (**f**) plotted versus the central g-force used during size selection. The dashed line in e illustrates the expected scaling. The central g-force is the mean of the upper and lower g-force boundaries used to prepare each fraction. **g**, **h** Mean nanosheet width (**g**) and thickness (**h**), each plotted versus mean nanosheet length. The lines show extreme values of nanosheet length/width aspect ratios (**g**) and length/thickness aspect ratios (**h**). In all cases the statistical data set from which histograms and means were calculated has a sample size of 120

simplicity we generally label them in order of decreasing size as XL, L, M, S, XS, XXS etc.

For each material, atomic force microscopy (AFM) was performed on the size-selected dispersions of each material with representative images of the XL and S sizes shown in Fig. 2 (Supplementary Figs. 2–20 for all). In all instances the nanosheets were reasonably well-exfoliated for all sizes and there was an obvious decrease in average size going from XL to XS. Nanosheet lengths (L, i.e. the longest dimension), widths (W, the dimension perpendicular to the length) and thicknesses (t, calibrated by step height analysis[2,29], see Methods) were measured for ~ 100-150 nanosheets in each fraction with sample histograms shown in Fig. 3a–d (see Supplementary Figs. 8–19 for all). Shown in Fig. 3e is a plot of the mean length $\langle L \rangle$ versus the central g-value for each faction and each material. Depending on the fraction/material, nanosheet lengths from >1000 nm to <50 nm were observed. In all cases, $\langle L \rangle$ scales inversely with the square root of central g-value (dashed line) in line with previous observations[3] and simple

models[31]. Shown in Fig. 3f is a plot of the mean nanosheet thickness $\langle t \rangle$ versus the central g-value showing scaling similar to $\langle L \rangle$. Thicknesses from ~1–40 nm were observed. We used this well-defined scaling with central g-force to somewhat reduce the number of samples for AFM statistics with the size of some intermediate fractions being estimated by interpolation (see Supplementary Figs. 8–20).

For LS studies on small particles, the deviation from spherical symmetry is important, as scattering from highly non-spherical particles is poorly understood. Shown in Fig. 3g, h are plots of nanosheet width (g) and thickness (h) versus length. Figure 3g shows most of the particles to have $\langle L \rangle / \langle w \rangle \approx 1.5$ with the smallest observed value of $\langle L \rangle / \langle w \rangle = 1.2$. However due to their belt-like nature, the $Cu(OH)_2$ nanosheets display values up to $\langle L \rangle / \langle w \rangle = 7$. Most relevant are the $\langle L \rangle / \langle t \rangle$ aspect ratios which can be extracted from Fig. 3h. This confirms all nanosheets to be platelet-like with length/thickness aspect ratios varying from $\langle L \rangle / \langle t \rangle = 9$ to 120. This shows that none of these samples can be

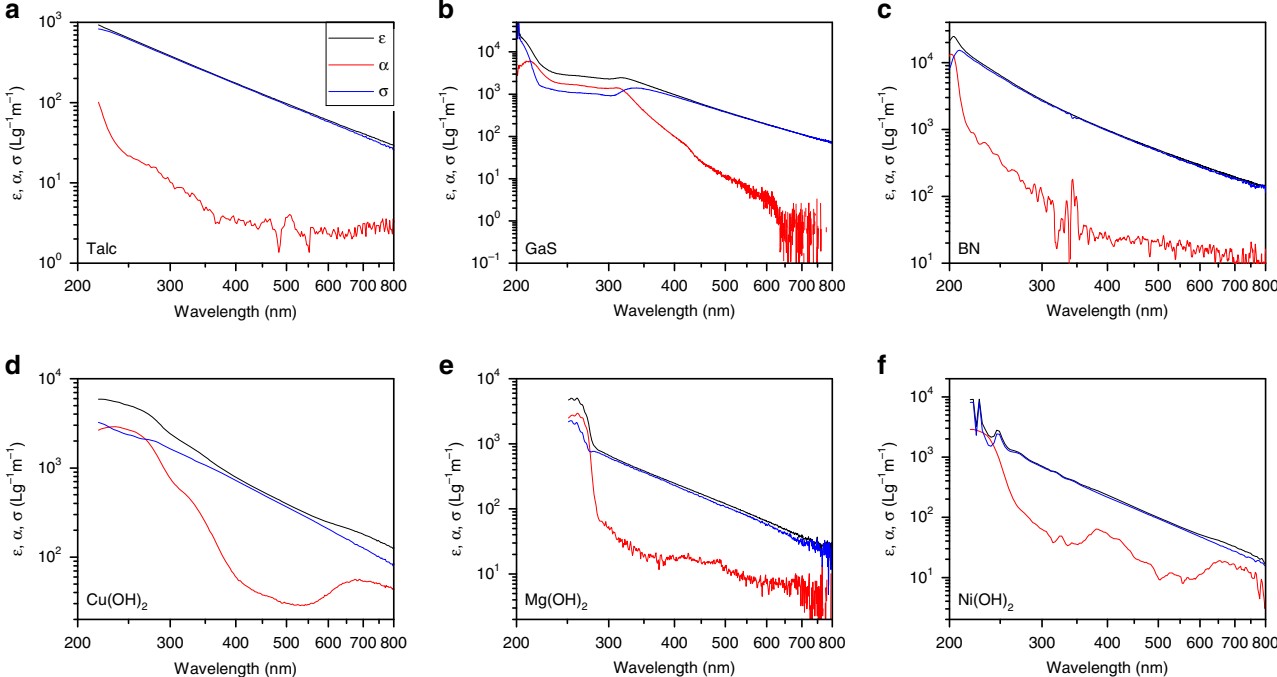

**Fig. 4** Optical coefficient spectra for size-selected nanosheets of each material. Each plot shows optical extinction ($\varepsilon$, black), absorption ($\alpha$, red) and scattering ($\sigma$, blue) coefficient spectra. In each case, the spectra are shown for the small (S) fraction

considered quasi-spherical with most being quasi-disk-like ($\langle L \rangle / \langle w \rangle \sim 1$ and $\langle L \rangle / \langle t \rangle \gg 1$) and $Cu(OH)_2$ being belt-like ($\langle L \rangle / \langle w \rangle \gg 1$ and $\langle L \rangle / \langle t \rangle \gg 1$).

**Size-dependent optical spectra**. In line with previous reports, we would expect the optical properties of nanosheets to depend sensitively on size[2,3]. Here, we used a standard optical spectrometer coupled with an integrating sphere (Methods) to measure the extinction, $\varepsilon(\lambda)$, scattering, $\sigma(\lambda)$, and absorption, $\alpha(\lambda)$, coefficient spectra for dispersions of all size-fractions of each 2D material in aqueous surfactant solutions (and in one case, GaS, the solvent NMP). We note that neither the surfactant solution nor solvent displayed any observable absorption or scattering in the spectral region of interest. Sample spectra from fraction S are shown in Fig. 4 (Supplementary Fig. 21 for all). Note that they are plotted on a log scale to reveal features in the absorbance. In all cases, the extinction spectra are relatively featureless and largely power-law-like as has previously been observed for liquid-exfoliated BN, GaS and $Ni(OH)_2$ nanosheets[5,6,16]. Such spectra are almost always erroneously presented in papers on 2D materials as "absorption spectra"[16].

However, the true absorption spectra are very different from the extinction spectra, appearing as one would expect for semiconductors with well-defined absorption onsets. In addition, they display richer structure as well as lower coefficients compared to the extinction spectra. For example, the absence of scattering allows the optical gap to be clearly observed at wavelengths of 250–400 nm (3.1–4.9 eV) depending on the material. In addition, the absorption spectra for some of the hydroxides are quite interesting as they displays features associated with the metal ion within the hydroxide[5].

Because the absorption coefficients are low, the scattering coefficient spectra dominate the extinction spectra. This is clear from Fig. 4 where the extinction and scattering spectra generally overlap, especially at long-wavelength where absorption is weak. To examine them more clearly, we plot the scattering coefficient spectra for each size-selected fraction of each material in Fig. 5. (NB: All scattering coefficient spectra were invariable with nanosheet concentration (Supplementary Fig. 22) and solvent (Supplementary Fig. 23) indicating multiple scattering to be negligible.) From Fig. 5, it is clear that the long-wavelength scattering coefficient values tend to increase with increasing nanosheet size. In all cases, at higher wavelengths, the spectra display well-defined power-law behaviour as previously observed for nanosheet dispersions[2,5,6,16]. For shorter wavelengths, where the absorption coefficient becomes non-negligible, $\sigma(\lambda)$ tends to deviate from pure power-law behaviour as has been observed previously for dispersions of $MoS_2$ nanosheets[2].

It is this power-law behaviour in the non-resonant regime that is the focus of the paper. We can describe it quantitatively using the expression $\sigma(\lambda) = K(\lambda/\lambda_0)^{-m}$, where for convenience we set $\lambda_0 = 1$ µm, meaning $K$ is the scattering coefficient at $\lambda = \lambda_0 = 1$ µm. (N.B.: We choose the arbitrary value of 1 µm to illustrate how the scattering coefficient varies with nanosheet size at fixed wavelength in the non-resonant regime.) Such power-law behaviour is stereotypical of light scattering[10–12], with the most well-known example being Rayleigh scattering which displays $m = 4$[14]. However, a number of papers have observed LS consistent with $m < 4$, typically for relatively large scattering particles[10–12]. This is particularly true for nanosheet dispersions where values of $m$ in the range of ~2–4 have been reported[5,6,17–20].

**Applying scattering approximations in the non-resonant regime**. In general, the scattering of light by small particles or molecules is described by Mie theory[14]. The more well-known Rayleigh scattering is an approximation to Mie scattering applicable for spheres with diameter $D < \lambda/10$. Here we are interested in the wavelength-dependence of the scattering coefficient. Although Eq. (3) represents the standard expression for Rayleigh scattering from spheres, for reasons that will become clear below, we will use an approximation quoted by van de

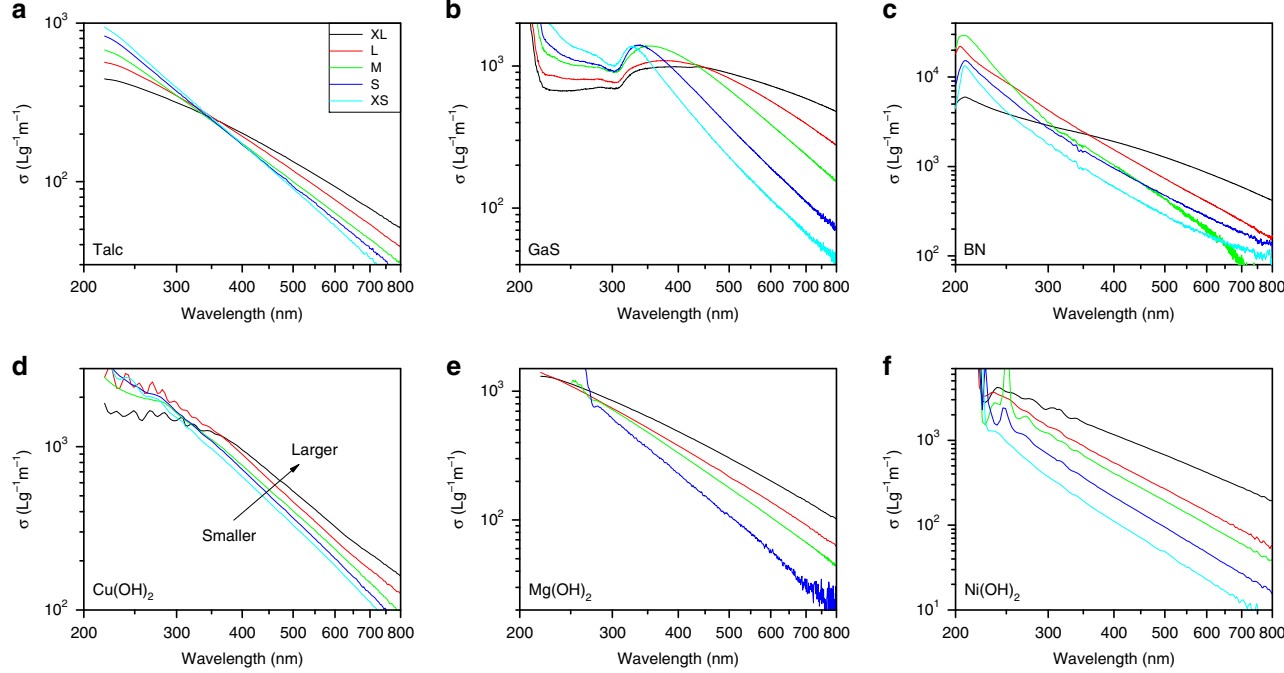

**Fig. 5** Scattering coefficient ($\sigma$) spectra for size-selected fractions for each material. Each panel shows data for five fractions, labelled XL, L, M, S, XS with nanosheet size decreasing from XL to XS (although for some materials, additional fractions were prepared). The colour scheme is the same in all panels

Hulst[14] (Supplementary Note 2) which is reasonably accurate for $1 \leq n/n_0 \leq 1.5$, as is the case here, and yields an expression for the scattering coefficient:

$$\sigma(\lambda) = \frac{16\pi^4 (n/n_0 - 1)^2}{9\rho D} \left(\frac{\lambda}{D}\right)^{-4} \quad (4)$$

where $\rho$ is the mass density of the spheres. We note that a very similar functional form (i.e. $\sigma(\lambda) \propto (n/n_0 - 1)^2 \lambda^{-4}$) also applies to discs which are small relative to the wavelength of light[14,32]. (N.B. in the non-resonant regime where $\alpha$ is generally very small, $\sigma$ [L/g/m] is defined via the transmittance by $T \cong 10^{-\sigma Cl}$ where $C$ is the dispersed particle concentration [g/L] and $l$ is the cell length [m]).

In the Rayleigh scattering regime, the scattering coefficient scales with $\lambda^{-4}$ (Eqs. 3 and 4). To illustrate this, in Fig. 6a, we have plotted $\sigma(\lambda)$ calculated using the Rayleigh approximation (Eq. 4) for spheres ($D = 125$ nm) using $n = 1.76$, $n_0 = 1.33$, $\rho = 4100$ kg/m$^3$ (mimicking Ni(OH)$_2$ in water/surfactant). The yellow field shows the region ($\lambda > 10D$) where Eq. (4) applies. Clearly, for spheres of $D \sim$ a few 100 nm, this wavelength range does not overlap the visible spectrum.

A reasonably good approximation to Mie theory exists for non-absorbing spheres with $n/n_0 < 2$ and diameter $D > \lambda/\pi$. Termed the van de Hulst (vdH) approximation (or the anomalous diffraction approximation), this is usually stated in terms of the scattering efficiency, $Q$, which is defined as the ratio of the scattering cross section ($\sigma_{CS}$) to the geometric cross section ($\pi D^2/4$)[14]:

$$Q = 2 - \frac{4}{p}\sin p + \frac{4}{p^2}(1 - \cos p) \quad (5)$$

where $p = 2\pi D(n/n_0 - 1)/\lambda$.

Converting scattering cross section to the scattering coefficient ($\sigma = \sigma_{CS}/(\rho V)_{sphere}$) associated with a dispersion of such spheres

yields

$$\sigma(\lambda) = \frac{3}{2}\frac{Q(\lambda)}{\rho D} \quad (6)$$

Using these equations, we plotted $\sigma(\lambda)$, calculated using the vdH approximation, versus $\lambda$ in Fig. 6a (taking $n = 1.76$, $n_0 = 1.33$ and $\rho = 4100$ kg/m$^3$ and $D = 125$ nm as before). The resultant spectrum is shown by the solid blue line and oscillates about some limiting value for $\lambda < D/2$, but follows a power-law for $\lambda > D/2$. The pink field shows the maximum range where the vdH approximation is expected to be appropriate ($\lambda < \pi D$). Similar to Rayleigh scattering, for spheres of $D \sim$ a few 100 nm, this wavelength range hardly overlaps the visible spectrum.

Clearly, the Rayleigh and vdH regimes do not overlap[33]. Furthermore, Fig. 6a suggests that, for these specific nanosheets, neither the Rayleigh nor vdH approximations are valid over much of the usual experimental spectral range ($\lambda = 300$–900 nm). We can generalise this by noting that the limits of applicability of the Rayleigh and vdH approximations can be rearranged to show that neither approximation applies when $\pi < \lambda/D < 10$. Given that nanosheets produced by LPE typical have lateral sizes in the range ~50–500 nm and as non-resonant scattering is typically measured over $\lambda = 300$–900 nm means that $\lambda/D$ is typically in the range ~0.6 to ~18 for nanosheet dispersions. That these $\lambda/D$ ranges broadly overlap means that typical non-resonant scattering spectra of LPE nanosheets will typically fall in the no-mans-land between the Rayleigh and vdH approximations. However, these approximations do yield some insights which facilitate analysis of scattering spectra.

Shown in Fig. 6b is a scattering coefficient spectrum for the Ni(OH)$_2$: L sample ($\langle L \rangle = 128$ nm, close to the condition used to plot the theory approximations) with the Rayleigh and vdH approximations reproduced for comparison. A number of points can be noted. Firstly, the experimental data sits very close to both theoretical lines while agreeing perfectly with neither. Secondly,

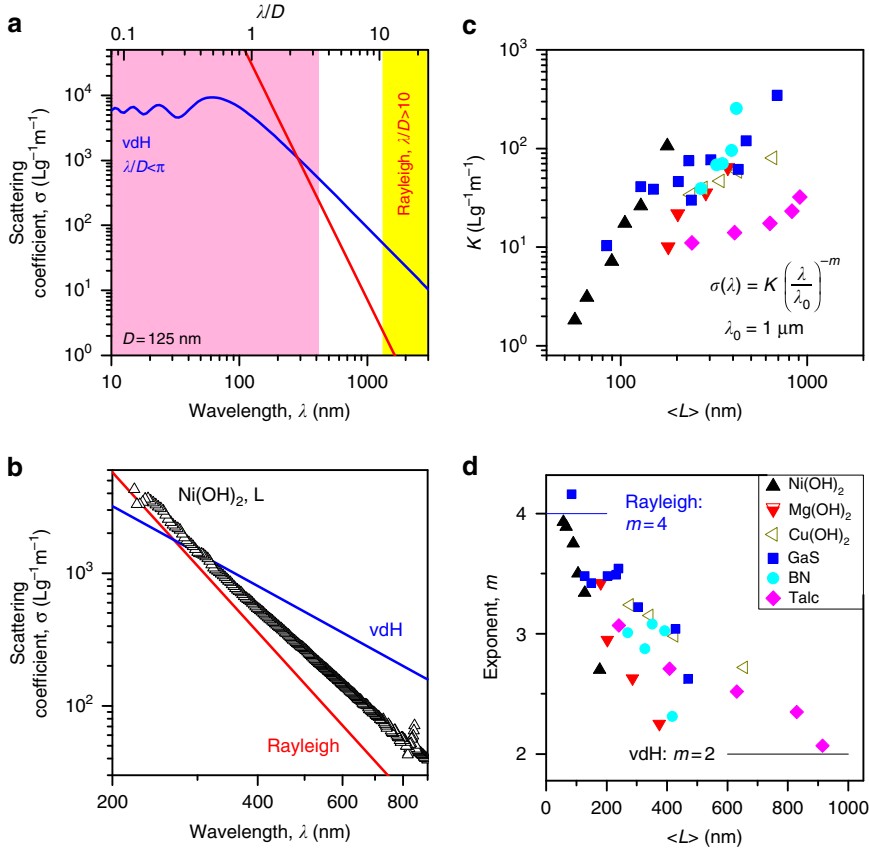

**Fig. 6** Light scattering in the non-resonant regime. **a** Theoretical scattering coefficients plotted versus wavelength (bottom axis) and $\lambda/D$ (top axis). The blue and red lines represents the van de Hulst and Rayleigh approximations for spheres ($D = 125$ nm) with $n = 1.76$, $\rho = 4100$ kg/m$^3$ (appropriate for Ni (OH)$_2$). The pink and yellow fields show the approximate ranges where we expect the vdH and Rayleigh approximations to be accurate. **b** Measured scattering coefficients plotted versus wavelength for the Ni(OH)$_2$:L sample ($\langle L \rangle = 128$ nm) with the approximations mentioned above plotted for comparison (taking $D = \langle L \rangle = 128$ nm). **c** Scattering parameter, $K$, plotted versus $\langle L \rangle$ for all samples (defined via $\sigma = K[\lambda/\mu m]^{-m}$). (N.B.: When defined in this way, $K$ represents the scattering coefficient at $\lambda = 1$ μm.) **d** Scattering exponent, m, plotted versus mean nanosheet length. The solid lines show the values expected from the Rayleigh and vdH approximations. The legend in **d** also applies to **c**

in this wavelength range, both theoretical curves are power-law-like, as is the experimental data. Interestingly, the slope (i.e. the power-law exponent) associated with the experimental power-law appears to be between the slopes of the Rayleigh (exponent −4) and vdH (exponent −2) approximations. As we will show below, these effects are due to the fact that LS by liquid-exfoliated nanosheets tends to fall between the limits set by Rayleigh and vdH scattering.

While it is obvious that Rayleigh scattering should be power-law-like in $\lambda$, it is less clear that this should be the case for the vdH approximation. We can clarify this by expanding Eq. (5) in the limit of small $p$ (i.e. large $\lambda$), obtaining $Q \approx p^2/2$ which can be converted to scattering exponent (via scattering cross section) yielding

$$\sigma(\lambda) \approx \frac{3\pi^2(n/n_0 - 1)^2}{\rho D}\left(\frac{\lambda}{D}\right)^{-2} \quad (7)$$

This equation shows that the vdH approximation can be power-law-like and applies over the wavelength range where the vdH approximation holds and where the expansion used above is valid (approximately $2\pi D(n/n_0 - 1) < \lambda < \pi D$). This equation is consistent with the observed power-law scaling ($\sigma(\lambda) = K(\lambda/\lambda_0)^{-m}$), and implies that, for larger spheres, the scattering exponent is $m = 2$.

While the vdH approximation, expressed via Eq. (7), applies to large spheres, there is good reason to believe that it provides a reasonable starting point to study scattering from 2D particles. It has been shown that, for large discs (diameter≫$\lambda$) oriented perpendicular to the incident light, the scattering coefficient scales as $\sigma(\lambda) \propto (n/n_0 - 1)^2 \lambda^{-2}$ (Supplementary Note 2)[14]. It is reasonable to assume that this general form may be orientation independent.

The discussion above implies that nanosheets display power-law scaling, similar to that expected for spheres, with exponents between $m = 4$ (Rayleigh scattering) and $m = 2$ (vdH behaviour). To test this, we fitted the high-wavelength portion of all scattering coefficient spectra to empirical power-laws ($\sigma(\lambda) = K(\lambda/\lambda_0)^{-m}$), finding good fits in all cases (Supplementary Fig. 24 shows examples). The parameters extracted from the fits ($K$ and $m$, where $\lambda_0 = 1$ μm) are plotted versus $\langle L \rangle$ in Fig. 6c, d. (N.B.: We expect power-law behaviour only at high wavelength for two reasons: at low wavelengths either the vdH approximation becomes non-power-law-like or, as $\alpha$ becomes non-zero, the non-resonant condition inherent in the derivation of both Eqs. (4) and (7) is broken.)

The strength of the LS can be represented by $K$ (i.e. the scattering coefficient at $\lambda = 1$ μm). Shown in Fig. 6c are values of $K$, plotted versus $\langle L \rangle$ for all six materials. In all cases, $K$ increases with $\langle L \rangle$ indicating that larger nanosheets are stronger scatterers, in line with

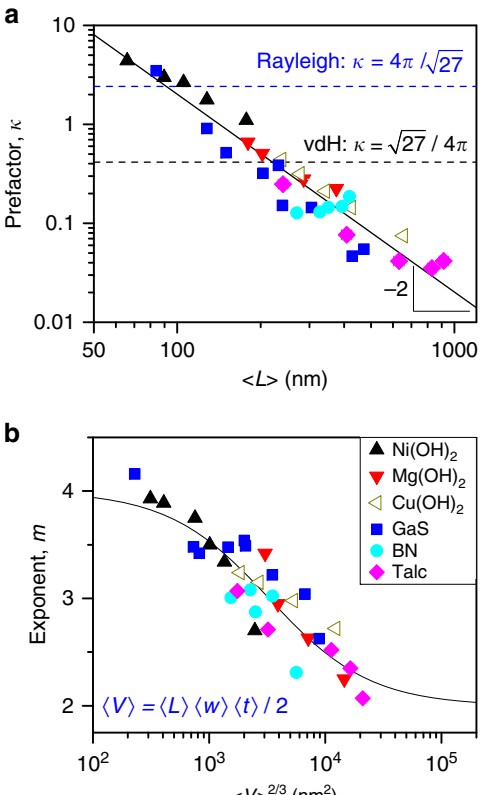

**Fig. 7** Light scattering master-curves. **a** Scattering pre-factor, $\kappa$, plotted versus mean nanosheet length. The solid line shows an empirical power-law fit while the dashed lines represents the high-$\lambda$ expansion of the van de Hulst equation (Eq. 7) and the Rayleigh approximation (Eq. 4), both with $D$ replaced by $\langle L \rangle$. **b** Scattering exponent plotted versus characteristic nanosheet area ($\langle V \rangle^{2/3}$) showing approximate master-curve behaviour. The line shows an empirical fit. The legend in **b** applies to **a** and **b**

the data in Fig. 5. However, there is considerable variation among the materials with no apparent pattern discernible.

The values of $m$ extracted from the fits are plotted against $\langle L \rangle$ for the six materials under study in Fig. 6d. This graph shows that for the largest nanosheets, the measured values of $m$ approach 2, similar to the vdH prediction for large spheres (lower line). However, as the nanosheet size was reduced, the exponent increased smoothly, appearing to approach $m = 4$ for extremely small nanosheets, as might be expected from the Rayleigh scattering approximation (upper line). This suggests that in this size/wavelength range, we are indeed seeing a transition from Rayleigh to vdH behaviour with increasing nanosheet size. It is worth noting that calculations by Granovskii et al. have predicted scattering in systems where $\lambda/D$ is intermediate between the Rayleigh and vdH regimes displays scattering coefficients which scale as $\lambda^{-m}$ with $2 < m < 4$[33].

**An empirical generalisation of the Rayleigh and vdH approximations**. Analysing visible-IR LS from LPE nanosheets faces two problems as described above. Firstly, they display disk-like geometry whereas the simple scattering approximations are appropriate to spheres. Secondly, such nanosheets tend to be produced in sizes which fall between the ranges covered by the Rayleigh and vdH approximations in the visible regime. In order to apply quantitative data analysis, we have developed an empirical equation, based on Eqs. (4) and (7), to model the experimental

nanosheet scattering data:

$$\sigma(\lambda) = \kappa \frac{4\pi^3}{\sqrt{3}} \frac{(n/n_0 - 1)^2}{\rho \langle L \rangle} \left( \frac{\lambda}{\langle L \rangle} \right)^{-m} \quad (8)$$

This equation is essentially a generalisation of Eqs. (4) and (7) and exploits the similarities between them while accounting for the differences. Equation (8) describes the transition region between Rayleigh and vdH scattering regimes while also capturing the differences between scattering from discs and spheres. To achieve this, we make three main modifications to Eqs (4) and (7). Firstly, Eq. (8) is written in terms of the mean nanosheet length, $\langle L \rangle$, to reflect the fact that Mie scattering of high-aspect-ratio objects should be sensitive to the longest dimension as this should give the largest phase difference between scattered waves. In addition, we include an arbitrary exponent, $m$, consistent with the experimental observation that $m$ is not necessarily equal to either 2 or 4. Finally, we include a scattering pre-factor $\kappa$ to represent deviations in scattering strength from that predicted by either model. We note that $\kappa$ is a measure of the differences between measured, non-resonant scattering for platelets versus that predicted for either small (Rayleigh) of large (vdH) spheres and may depend on the nanosheet size. In addition, we write the numerical pre-factor $(4\pi^3/\sqrt{3})$ as the geometric mean between Rayleigh $(16\pi^4/9)$ and vdH $(3\pi^2)$ prefactors. We note that both Rayleigh and vdH expressions are recovered by writing $\kappa = 4\pi/\sqrt{27}$ or $\kappa = \sqrt{27}/4\pi$, respectively. Unlike $K$, we would expect $\kappa$ to be material independent with all materials-specific information encoded in $n$ and $\rho$.

We can use Eq. (8) to extract the scattering strength, $\kappa$, from the $K$- and $m$-data in Fig. 6c, d, taking $n_0$ as appropriate and using values of $n$ extracted from the literature (Table 1). These $\kappa$ data are plotted versus mean nanosheet length in Fig. 7a. We find the remarkable fact that $\kappa$-data for all six materials fall very close to the same master-curve which is characterised by a power-law scaling with $\langle L \rangle$, $\kappa = [L_0/\langle L \rangle]^2$, where $L_0$ is an empirical parameter equal to 142 nm ($R^2 = 0.92$). Within this framework, the parameter $L_0$ represents the nanosheet length where $\kappa = 1$ and so describes a situation which is midway between Rayleigh and vdH scattering. We note that this behaviour deviates significantly from that predicted by both the vdH and Rayleigh approximations (dashed lines). This master-curve-like behaviour is not unexpected as, in the framework of the vdH approximation, $\kappa$ includes no material parameters. However, it is reassuring that such behaviour is found experimentally. Clearly, incorporating the empirical expression for $\kappa$ in Eq. (8) yields our final equation for the scattering coefficient

$$\sigma(\lambda) = \frac{4\pi^3}{\sqrt{3}} \frac{(n/n_0 - 1)^2 L_0^2}{\rho \langle L \rangle^3} \left( \frac{\lambda}{\langle L \rangle} \right)^{-m} \quad (9)$$

Encouraged by the finding that the $\kappa$ versus $\langle L \rangle$ data fell on a master-curve, we explored the possibility that the $m$-data might also fall on a master-curve when plotted versus the appropriate dimensional parameter. To do this, we used trial and error to search for master-curve-behaviour by plotting $m$ versus a range of dimensional parameters such as mean nanosheet length, width, thickness, aspect ratio etc as described in the SI (Supplementary Fig. 25 and Supplementary Note 3). As shown in Fig. 7b, we found a reasonable master-curve when the exponent, $m$, was plotted versus the mean nanosheet volume, $\langle V \rangle$, raised to the power of 2/3 (the mean volume is calculated from $\langle V \rangle = \langle L \rangle \langle W \rangle \langle t \rangle/2$, see Supplementary Note 3 for justification). This parameter,

$\langle V \rangle^{2/3}$, would be expected to influence the scattering of light as it represents a characteristic area associated with the nanosheet and is a measure of the average nanosheet area presented to the incident light. Empirically (see Supplementary Figure 25), we found that the master-curve could be fitted to the sigmoidal expression $m = a + (4 - a)/[1 + (\langle V \rangle / V_0)^{2/3}]$ yielding $a = 2.07 \pm 0.15$. This supports the idea that in the large-nanosheet limit, $m = 2$. As a result, we propose the $m$ versus $\langle V \rangle$ data can be described by

$$m = 2 + 2/[1 + (\langle V \rangle / V_0)^{2/3}], \quad (10)$$

where fitting ($R^2 = 0.80$) yields $V_0 = 1.88 \times 10^5$ nm$^3$ (Fig. 7b).

This result is consistent with the idea that for very large nanosheets, $m \to 2$ as per the van der Hulst approximation and with the expectation from Rayleigh scattering that $m \to 4$ for small nanosheets. Importantly, data for liquid-exfoliated nanosheets lie between these limits with Fig. 7b showing a smooth transition from Rayleigh scattering ($m = 4$) to vdH-type behaviour ($m = 2$) as nanosheet size increases.

It is worth noting that the master-curves in Fig. 7 apply only to disk-like particles. While similar experiments, performed on spherical nano-particles, yielded qualitatively similar results, the data did not fall on the master-curves reported above (Supplementary Fig. 26).

Equations (8) and (9) represent a semi-empirical combination of Rayleigh and van der Hulst scattering approximations. Combined with Eq. (10), they describe the transition from small ($m = 4$) to large ($m = 2$) nanosheet behaviour. We note that for very small flakes $\kappa$ is quite close to the Rayleigh prediction (Fig. 7a), probably because the shape of very small objects should not affect scattering dramatically. However, for large nanosheets, measured values of $\kappa$ deviate significantly from the vdH approximation. We believe this a manifestation of the differences in scattering by large disc-like particles compared to that from large spheres on which the vdH model is based.

**Onset of scattering**. We can use the model outlined above to estimate the nanosheet size below which scattering can be ignored. To do this, we note that scattering is always more intense at low wavelength. As such, we can define the onset of scattering to occur for a specific nanosheet length, $\langle L \rangle_{\text{Onset}}$, such that the scattering and absorption coefficients are equal at a wavelength, $\lambda_{\text{NR-max}}$, representing the highest energy within the non-resonant regime, i.e. $\alpha_{\text{NR-max}} = \sigma_{\text{NR-max}}$. For nanosheets above this size, the increased scattering coefficient will result in a measurable extinction in the non-resonant regime. Thus the onset of scattering will occur when

$$\alpha_{\text{NR-max}} = \sigma_{\text{NR-max}} = \frac{L_0^2}{\langle L \rangle_{\text{Onset}}^2} \frac{4\pi^3}{\sqrt{3}} \frac{(n/n_0 - 1)^2}{\rho \langle L \rangle_{\text{Onset}}} \left( \frac{\lambda_{\text{NR-max}}}{\langle L \rangle_{\text{Onset}}} \right)^{-m} \quad (11)$$

Assuming that this occurs for small nanosheets where we can approximate $m = 4$, we can rearrange this equation to get:

$$\langle L \rangle_{\text{Onset}} = \frac{\sqrt{3}}{4\pi^3 L_0^2} \alpha_{\text{NR-max}} \lambda_{\text{NR-max}}^4 \frac{\rho}{(n/n_0 - 1)^2} \quad (12)$$

Clearly, the onset of scattering depends on nanosheet type. Taking GaS in NMP as an example, Supplementary Fig. 21 shows $\lambda_{\text{NR-max}} \approx 450$ nm and $\alpha_{\text{NR-max}} \approx 100$ L/g/m. Then, using the density

and refractive values given in the Supplementary Table 1, we find $\langle L \rangle_{\text{Onset}} \sim 30$ nm, broadly in line with experience.

## Discussion

The data described above show that dispersions of randomly oriented 2D platelets scatter light in a manner which resembles that expected for spheres. However, we find important deviations from the predictions of simple sphere-based models. Firstly, as nanosheet volume increases, the power-law scattering exponent displays a gradual transition from the small-sphere value of 4 to the large-sphere value of 2. This is a manifestation of the fact that liquid-exfoliated nanosheets usually have lateral sizes between the limits associated with the Rayleigh and van der Hulst sphere-based approximations. This behaviour allows the estimation of platelet volume from the scattering exponent which can be used as a universal spectroscopic metric, for example to monitor nanosheet aggregation (see Supplementary Note 4, 27–29). Secondly, we find that the scattering intensity is different from the predictions of both sphere-based approximations. However, these approximations can be modified by including a pre-factor which displays universal scaling with length for all nanosheets studied. This pre-factor corrects the scattering models both for the effects of shape (nanosheets are not spherical) and the fact that their size lies outside the ranges covered by either approximations. In addition, its universal scaling means the pre-factor can be used estimate mean lateral nanosheet sizes from the scattering coefficient spectra of dispersions of any wide-bandgap nanosheet (see Supplementary Note 4, 30–31).

## Methods

**Materials**. β-Nickel hydroxide powder (>95% 283622), magnesium hydroxide (95% 310093), copper hydroxide (289787), talc (243604), boron nitride (255475) and sodium cholate were purchased from Sigma Aldrich. gallium sulphide was purchased from American Elements (99.999% GaS-05-P). De-ionized water was prepared in house and all solvents used were purchased with the highest available purity.

**Preparing nanosheet dispersions**. For the hydroxides, 1.6 g of powder was pre-treated by sonicating using a sonic tip in de-ionized 80 mL deionised water for 1 h. The dispersion was then centrifuged at 4.5 krpm (2150 × g) for 1 h and decanted with the sediment being retained and dried. The pre-treated material (20 g/L) was then sonicated in 9 g/L of sodium cholate and de-ionized water solution using a flat head tip (Sonics VX-750) with 60% amplitude and 6 s on/2 s off for 4 h. To provide necessary temperature control, ice cooling was used. Once sonicated, the dispersions were centrifuged and size-selected by LCC. BN was exfoliated in water/sodium cholate according to a previously published method[26]. GaS was also exfoliated in NMP according to a previously published method[6]

**Size selection**. Liquid cascade centrifugation was used with subsequently increasing rotation speeds as previously reported[17]. In the case of the hydroxide dispersions, the stock obtained after sonication was centrifuged at 25 × g for 60 min in a Hettich Mikro 220R with a fixed angle rotor 1016. The sediment was discarded and the supernatant was centrifuged at 100 × g for 60 min. The sediment after this centrifugation step was redispersed in fresh surfactant solution (1 h bath sonication, $c_{\text{SC}} = 9$ g/L) producing the largest size. The supernatant after the 100 × g centrifugation step was centrifuged at 250 × g for 60 min, producing the second largest size in the redispersed sediment. These steps were repeated in further increments of 400 × g, 1000 × g, and 3000 × g, thus producing five sizes. The BN dispersions were size-selected with different centrifugal accelerations in analogy to a previously reported method[26]. For GaS, each centrifugation run was carried out for 2 h due to the higher viscosity of the solvent. In addition, a centrifugation run at 10,000 × g (using a 1195-A rotor) was carried out to have access to smaller/thinner nanosheets. In addition, two samples were centrifuged at 1/10th of the initial lower centrifugation boundary for 14 h in an attempt to decouple the length and thickness relationship. Details are given in the SI. To facilitate characterisation and deposition, all GaS sediments were redispersed in isopropanol by 2 min bath sonication.

**Characterization and equipment**. Optical extinction and absorbance was measured on a PerkinElmer 950 spectrometer in quartz cuvettes with a path length of 0.4 cm. To distinguish between contributions from scattering and absorbance to the

extinction spectra, dispersions were measured in an integrating sphere using a sample holder to place the cuvette in the centre of the sphere. (N.B.: Cuvettes need to be transparent to all sides and correct positioning is important.) The absorbance spectrum is obtained from the measurement inside the sphere. A second measurement on each dispersion was performed outside the sphere in the standard configuration to obtain the extinction spectrum. This allows calculation of the scattering spectrum (extinction minus absorbance). In general, all optical measurements were made at relatively low concentrations (~0.02–0.1 mg/ml). As a result, the maximum extinction associated with the nanosheets (in the resonant regime) was ~0.5 and always much lower in the non-resonant regime. (N.B.: spectra were measured only up to 800 nm even though the spectrometer is capable of reaching 1400 nm. This is because a lamp-change just beyond 800 nm tends to introduce artefacts which hamper analysis of the scattering spectra.)

Low-resolution bright field transmission electron microscopy imaging was performed using a JEOL 2100, operated at 200 kV. Holey carbon grids (400 mesh) were purchased from Agar Scientific and prepared by diluting a dispersion to a low concentration and drop casting onto a grid placed on a filter membrane to wick away excess solvent. Statistical analysis was performed of the flake dimensions by measuring the longest axis of the nanosheet and assigning it as "length", $L$ and the perpendicular to the longest axis and assigning it as "width", $W$.

Atomic force microscopy (AFM) was carried out on a Dimension ICON3 scanning probe microscope (Bruker AXS S.A.S.) in ScanAsyst in air under ambient conditions using aluminium coated silicon cantilevers (OLTESP-R3). The concentrated dispersions were diluted with water to yield a transparent dispersion. A drop of the dilute dispersions (20 μL) was deposited on a pre-heated (180 °C) Si/SiO$_2$ wafer (1 × 1 cm$^2$) with an oxide layer of 190 nm. Immediately after deposition, the wafers were rinsed with ~5 mL of water and ~5 mL of isopropanol. Typical image sizes were 10 × 10 μm$^2$ at scan rates of 0.5 Hz with 1024 lines per image. Length, width and thickness was measured for ~250 individually deposited nanosheets. To correct the nanosheet length due to tip broadening, we used a previously established length correction[34]. To convert thickness into layer number, step height analysis was used as demonstrated previously[2]. For Ni(OH)$_2$, GaS, BN and talc previously published step heights on liquid phase exfoliated nanosheets were used[5,6,25,26]. A value of 1.05 nm was used for Cu(OH)$_2$ and Mg(OH)$_2$ as this is the value for both Ni(OH)$_2$[5] and Co(OH)$_2$[35] and we assume that materials within one family have similar step heights. The as obtained layer-numbers were used to calculate the mean thickness ⟨$t$⟩ by multiplying the mean layer number by the theoretical thickness of one layer (0.5 nm for the hydroxides, 0.35 nm for BN, 0.4 nm for GaS and 1 nm for talc). It should be noted that AFM on hydroxides is challenging due to the relatively high surfactant concentration (9 g/L) required to maintain colloidal stability. To reduce the surfactant contamination on the wafer, the samples were nonetheless diluted with water (factor 30–200 depending on the concentration) and wafers cast immediately after dilution. Nonetheless, reaggregation on the wafer was observed frequently. Therefore, extreme care was taken, to only measure length, width and thickness of individually deposited nanosheets.

For Raman spectroscopy, the samples were centrifuged at 10,000 × $g$ and redispersed at high nanosheet concentration in water to reduce the surfactant concentration which otherwise contributes to the Raman spectra due to the non-resonant nature of the excitation. 10 μL of the dispersion was dropcast onto alumina foil and dried in air. Raman measurements were performed with a Renishaw InVia microscope with 532 nm excitation laser in air under ambient conditions. The Raman emission was collected by a 50×, long working distance objective lens in streamline mode and dispersed by a 2400 l/mm grating with 1% of the laser power (<0.2 mW). A map over 20 × 20 μm$^2$ was recorded with 100 spectra that were averaged and baseline corrected.

## Data availability

Data that supports the findings of this study are available from the corresponding author upon reasonable request.

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

## Acknowledgements

This work was primarily funded through the European Union Seventh Framework Programme under grant agreements 604391 and 785219 Graphene Flagship and the Science Foundation Ireland (SFI) funded centre AMBER (SFI/12/RC/2278).

## Author contributions

A.H. prepared samples, performed optical measurements and TEM. C.B. performed and analysed AFM and Raman, J.B.B., X.H. and C.G. performed optical measurements, A.G. and B.S. performed AFM, J.F.D. and J.N.C. developed the model, J.N.C. conceived the work and wrote the manuscript.

## Additional information

**Competing interests:** The authors declare no competing interests.

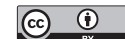

