## [Peer Review File · Nature Communications]

Reviewers' comments:

Reviewer #1 (Remarks to the Author):

In this work the authors explore the non-resonant light scattering properties of a number of wide-bandgap 2D disc-like nano-materials. They develop a model which allows the non-resonant scattering spectra to be used as metrics for particle size and aggregation in the nanosheet dispersion. Specifically, they found that the scattering coefficient scales as a power law with wavelength at the range of 300 nm – 900 nm lying roughly between the predictions of Rayleigh and van der Hulst scattering, with the power-law coefficient (m) transitioned from Rayleigh-like ($m=4$) to van der Hulst-like ($m=2$) as the nanosheet size is increased. Based on the two above approximations of the Mie theory they generate an empirical expression to calculate the scattering coefficients in the intermediate non-resonant regime of interest. By fitting this expression to the experimental long-wavelength power-law part of the scattering spectrum they can derive the mean nanosheet dimensional parameters.

The paper is well organized presenting step by step the theoretical analysis accompanied by a rich collection of experimental measurements to validate the construction of the model. The results will be of interest to the community spurring more discussion and enabling the size characterization of 2D high aspect-ratio nano-materials by simple optical extinction spectroscopy.

However, the proposed evaluation for the paper is major revision because there are several points that should be cleared and properly commented before it is recommended for publication:

1. The authors use the nano-sphere model to describe nano-disks. In the small particle limit this is justified and the authors could cite the relevant approximation for nano-disks (Bohren and Huffman "Absorption and scattering of light by small particles", pg 145-146), where the main change is $V_{\text{sphere}} * [(n/n_0)^2 - 1]^2 / [(n/n_0)^2 + 2]^2 \rightarrow V_{\text{disk}} * [(n/n_0)^2 - 1]^2 / 9$. This justifies using the same functional form for the prefactor (especially when $n/n_0 \sim 1$) and the $m=4$ exponent as the upper limit.

2. However, approximating the very large flakes as spheres in the vdH regime puzzles me: here light can really distinguish the actual shape details. With that in mind, what is the theoretical justification of using the vdH functional form for the prefactor and assuming $m=2$ as the lower limit? At the end of the day it boils down to a fit, and while the $m=2$ limit seems to work ok, the prefactor is completely off for the larger flakes. To me the latter invalidates the whole discussion regarding the vdH regime and points to a simpler picture/process where you start from the Rayleigh formulation and just fit the prefactor and the exponent.

3. In line 420, why should $\langle V \rangle^{1/3}$ be a characteristic length for the flakes? Given their flat 2D character, all samples should fall into the Rayleigh regime under this criterion. As a matter of fact, I do not understand the explanation given for Fig. 7C. In my reading of the figure, almost all samples are outside the validity range of vdH even for $\lambda=300\text{nm}$, while at $\lambda=800\text{nm}$ (the spectral range where you perform the fit to extract m) almost all samples are within the validity range of Rayleigh. So if this $\langle V \rangle^{1/3}$ criterion is correct, then all samples should exhibit a $m=4$ behavior. I think Fig. 7C does not support the conclusions reached before and does not offer a viable model.

4. In Fig. 7B the exponent m is plotted as a function of $\langle V \rangle^{2/3}$. From all the discussion thus far I would expect it to also be a function of $\langle L \rangle$ and L_0 . Is there a physical reason why the former is more appropriate than the latter?

5. As noted in line 442 it is $\langle V \rangle = \langle L \rangle \langle w \rangle \langle t \rangle / 2$. Is it possible to calculate $\langle V \rangle = \langle Lwt \rangle / 2$ for the distributions? Wouldn't that be more appropriate as average volume?

6. In line 328 the correct approximation is $Q=p^2/2$ (it seems the authors expanded the sine term up to p^3 and the cosine term up to p^2 . But since the sine term is divided by p and the cosine term is divided by p^2 , they should expand the cosine term up to p^4)
7. In line 381 the Rayleigh and vdH prefactors are inter-switched.
8. In line 477, what is meant by "trial and error"?
9. In line 300: it should be figure 6A
10. In line 436 you mean Eq. 5 and 6?

Reviewer #2 (Remarks to the Author):

Optical extinction spectroscopy is one important technique to characterize the properties of new molecules or nano-materials, and the extinction spectra contain contributions from both absorption and scattering. Scattering can dominate optical extinction spectra and cannot be ignored in some molecular solutions and suspensions of large nanoparticles in the range 50-1000 nm. Scattering can completely obscure absorption features hampering quantitative spectral analysis. However, many researchers are confused on extinction spectra and absorption spectra. As a result, it is quite necessary and important to distinguish these two spectra.

In this manuscript, Andrew Harvey et al. selected six different 2D materials with large bandgaps, produced each kind of nanosheets with five different sizes, and investigated the non-resonant light scattering in order to study the nanosheet parameters control the scattering coefficient spectra. They also afforded a simple model to estimate the magnitude of the scattering coefficient as a function of wavelength for different nano-materials of different sizes. They found that the scattering coefficient spectra lied roughly between the predictions of Rayleigh and van der Hulst, and with the nanosheet size increasing, the scattering power-law coefficient transitioned from Rayleigh-like to van der Hulst-like. This work affords a systematical instruction on non-resonant light scattering in nanosheet suspensions.

The work is of high novelty and well organized. I think the manuscript can be accepted for publication if the below comments can be addressed.

1. Line 100, please give the typical α value of molecular system, or afford references.
2. Please add some discussion on what relationship between the particle dimension D and the wavelength, the scattering cannot be ignored? Is it same or similar in different kind of materials?
3. Does scattering change with the light intensity? Please add some comments on this. And please afford the light intensity when you measure the optical extinction, absorption and scattering spectra.
4. In the "Applying Rayleigh and van der Hulst approximations to the non-resonant scattering regime" part, you sue an approximation of $n/n_0 \sim 1$ and get equation (2b). However, figure 6A is plotted using $n=1.76$ and $n_0=1.33$. The value of n is much larger than n_0 . Please add some comments on this. If you do not use the approximation, how about the result will be?
5. Line 300, I am afraid the authors made a mistake in the sentence "versus λ in figure 4A". It should be figure 6A.
6. From figure 6B, we can see neither the two theoretical curves agree well with the experimental data. Could you please give some explanations?
7. Line 339, the authors claimed that good fits were obtained. Would you please afford several fitting results?

Reviewer #3 (Remarks to the Author):

The article by Harvey et al proposes the use of non-resonant light scattering to estimate flake sizes of exfoliated 2D materials.

The authors have used large band gap 2D material flakes in water-surfactant suspension, with the exception of GaS where they used NMP.

The authors show that the scattering coefficient spectra of all the studied materials lie between Rayleigh and van der Hulst models, transition from the former to the latter as the nanosheet size increased following a mastercurve of power law coefficient vs nanosheet volume.

Although the article is rather long with 8 composite figures, it is written very well considering the complexity of the experimental analysis presented. Figures are also clear and legible.

However, I am not entirely convinced with the model and analysis presented and am unable to recommend the publication of the article in Nat Comms.

Below I have listed (in no particular order) some of my concerns that the authors may consider for future submissions:

* The authors consider scattering coefficient at 1 μm . Why exactly?

Also, does 1 μm not fall within the non resonant regimes for some of the commonly studied semiconducting 2D materials too? If yes, couldn't they also be considered in this study (excluding the part of the absorption spectra associated with material bandgap)? I also note that most of the measurements are carried out till 800 nm, probably due to absorption from solvents used. This requires a clear explanation.

* The authors state that "As a result, the presence of this scattering background can obscure or shift the positions of absorption peaks, making detailed analysis difficult "- It is unclear why a broad background could noticeably shift the position of absorption peaks.

* The authors state:

Line 50: "Scattering contributions manifest themselves as a broad background in the resonant regime which falls off with increasing wavelength in the non-resonant regime"

Line 79: "scattering can dominate extinction spectra, especially in the long wavelength non-resonant regime. "

I am not sure how both the above statements could be correct.

* In line 103, the authors state "However, even though equation 2 only holds for ...". Do the authors mean both 2a and 2b?

* The authors include quite a detailed background on how scattering spectra has/has not been considered for 2D materials, including that of graphene. What is not clearly understood is, why these would be any different from any nanoparticles.

The authors state that the mastercurves in Fig 7 works only for disk-like particles (line 431), not spherical particles (like PS beads). It is a little confusing because the nanosheets are considered to be diamond shaped (line 403) for equation (7). Is it not sufficiently close to a spherical geometry? More specifically, how does this model work for other nanoparticles? Why, for example, this model does not work with gold nanoparticles and polystyrene nanobeads? or even carbon nanotubes?

* It is not clear whether the authors have considered the effect of absorption from the surfactants and micelles and the scattering from surfactant micelles (apart from GaS)

* It is not clear why GaS is the only material to be exfoliated in NMP.

* All of the dispersions appear cloudy in Fig 1B. If this is indeed the case, it indicates very strong scattering in the visible. Are these the XL or XS dispersions?

* The authors state that "Such spectra are almost always erroneously presented in papers on 2D materials as "absorption spectra".³⁶⁻³⁸ " While the statement is fully correct, it might be prudent to point out that most of the research groups working on 1 and 2D materials have made this error over the past few years and only some research groups have started to correct this as they have access to spectrometers with integrating spheres. For example, does not "doi:10.1038/nnano.2008.215", one of the highest cited articles in liquid phase exfoliation from the same group also made this error? In my humble opinion, it is unfair to associate papers from some specific groups about this when almost everyone in the field made the same error in judgement in recent years.

* The authors state that "in the non-resonant regime where $\alpha \sim 0$ " (Line 280). Is this strictly true? In addition to a certain carrier concentration at the temperature where the experiment is carried out, what about edge effects? I note that edge related states in 2D nanosheets for linear optical absorption has been considered by several groups in recent years.

* The number of data points in Fig E,F,G,H seem inconsistent (e.g. 6 in E, G and 5 points in F, H) unless some data points are directly below other entries. Should't there be same number of data points as we are looking at the same dispersions and using the same AFM scans for height, width and thickness measurements?

* The data in Fig 7B could be fitted significantly better with a straight line. The current fit and the associated text explaining the transition from Rayleigh to vdH is not convincing. It is also unclear why the nanosheets are modelled as diamond-shaped (line 403).

* The authors state that "For typical nanosheets with $L / W \sim 1.5$ and $L / t \sim 30$ (figure 3), V_0 would represent nanosheets with $L \sim 260$ nm, $W \sim 175$ nm and $t \sim 9$ nm ". How accurate is this estimation from the measurements performed? An indication would be useful.

* Could the authors please explain the statement "We found all κ vs. mean nanosheet length, L , data to fall on a materials-independent master curve, characterised by $\kappa \propto L^{-2}$, rather than remaining constant as expected for spheres." Are the authors implying that κ would remain unchanged regardless of the size of spherical scatterers?

* Is there any information available on the accuracy of the spectrometer for the measured concentration at the measured wavelength range?

Others:

* Typos:

Line 165: "[ref5]"

Line 229: "aspect ratios"

* It would be useful to include a scale bar for Fig 2.

f

Manuscript NCOMMS-18-19555 Harvey et al

Response to reviewers

Reviewers' comments:

Reviewer #1 (Remarks to the Author):

In this work the authors explore the non-resonant light scattering properties of a number of wide-bandgap 2D disc-like nano-materials. They develop a model which allows the non-resonant scattering spectra to be used as metrics for particle size and aggregation in the nanosheet dispersion. Specifically, they found that the scattering coefficient scales as a power law with wavelength at the range of 300 nm – 900 nm lying roughly between the predictions of Rayleigh and van der Hulst scattering, with the power-law coefficient (m) transitioned from Rayleigh-like ($m=4$) to van der Hulst-like ($m=2$) as the nanosheet size is increased. Based on the two above approximations of the Mie theory they generate an empirical expression to calculate the scattering coefficients in the intermediate non-resonant regime of interest. By fitting this expression to the experimental long-wavelength power-law part of the scattering spectrum they can derive the mean nanosheet dimensional parameters.

The paper is well organized presenting step by step the theoretical analysis accompanied by a rich collection of experimental measurements to validate the construction of the model. The results will be of interest to the community spurring more discussion and enabling the size characterization of 2D high aspect-ratio nano-materials by simple optical extinction spectroscopy.

However, the proposed evaluation for the paper is major revision because there are several points that should be cleared and properly commented before it is recommended for publication:

1. The authors use the nano-sphere model to describe nano-disks. In the small particle limit this is justified and the authors could cite the relevant approximation for nano-disks (Bohren and Huffman “Absorption and scattering of light by small particles”, pg 145-146), where the main change is $V_{\text{sphere}} * [(n/n_0)^2 - 1]^2 / [(n/n_0)^2 + 2]^2 \rightarrow V_{\text{disk}} * [(n/n_0)^2 - 1]^2 / 9$. This justifies using the same functional form for the prefactor (especially when $n/n_0 \sim 1$) and the $m=4$ exponent as the upper limit.

We have added this citation, modifying the text to read:

“We note that a very similar functional form (i.e. $\sigma(\lambda) \propto (n/n_0 - 1)^2 \lambda^{-4}$) also applies to disks which are small relative to the wavelength of light.^{14, 37}”

2. However, approximating the very large flakes as spheres in the vdH regime puzzles me: here light can really distinguish the actual shape details. With that in mind, what is the theoretical justification of using the vdH functional form for the prefactor and assuming $m=2$

as the lower limit? At the end of the day it boils down to a fit, and while the $m=2$ limit seems to work ok, the prefactor is completely off for the larger flakes. To me the latter invalidates the whole discussion regarding the vdH regime and points to a simpler picture/process where you start from the Rayleigh formulation and just fit the prefactor and the exponent.

This is a good point which has spurred us to improve the paper. Take the exponent first. The question is - even though large spheres have $m=2$, does this really mean large discs should also have $m=2$?

It turns out that there is an approximate solution to Rayleigh-Gans theory for large discs perpendicular to the beam which predicts $m=2$. We believe this justifies using the vdH approximation and implies that $m=2$ is a sensible large-disk limit in randomly oriented systems. We have added the following text (and SI elaboration):

“While the vdH approximation, expressed via equation 2, applies to large spheres, there is good reason to believe that it provides a reasonable starting point to study scattering from disk-like particles. It has been shown that, for large disks (diameter $\gg \lambda$) oriented perpendicular to the incident light, the scattering coefficient scales as $\sigma(\lambda) \propto (n/n_0 - 1)^2 \lambda^{-2}$ (see SI).¹⁴ It is reasonable to assume that this general form may be orientation independent.”

In addition, I should note that the m vs $\langle V \rangle^{2/3}$ data was originally fit to the function

$$m = a + \frac{(4-a)}{1 + \langle V \rangle^{2/3} / b}$$

This fit returns $a=2.07 \pm 0.15$ supporting the value of 2 as the large-nanosheet limit to m . This is included in the SI.

Turning to the prefactor, we believe that the deviation from the vdH prediction for large nanosheets is simply a manifestation of the shape difference between the measured disk-like objects and the spheres on which the original model is based.

“Equations 5 a&b represents a semi-empirical combination of Rayleigh and van der Hulst scattering approximations. Combined with equation 6, they describe the transition from small ($m=4$) to large ($m=2$) nanosheet behaviour. We note that for very small flakes κ is quite close to the Rayleigh prediction (figure 7a), probably because the shape of very small objects should not affect scattering dramatically. However, for large nanosheets, measured values of κ deviate significantly from the vdH approximation. We believe this a manifestation of the differences in scattering by large disc-like particles compared to that from large spheres on which the vdH model is based.”

3. In line 420, why should $\langle V \rangle^{1/3}$ be a characteristic length for the flakes? Given their flat 2D character, all samples should fall into the Rayleigh regime under this criterion. As a matter of fact, I do not understand the explanation given for Fig. 7C. In my reading of the figure, almost all samples are outside the validity range of vdH even for $\lambda=300\text{nm}$, while at $\lambda=800\text{nm}$ (the spectral range where you perform the fit to extract m) almost all samples are within the validity range of Rayleigh. So if this $\langle V \rangle^{1/3}$ criterion is correct, then all samples should exhibit a $m=4$ behavior. I think Fig. 7C does not support the conclusions reached before and does not offer a viable model.

Actually, on reflection we agree with the referee. I had thought fig 7C would clarify things somewhat. Clearly it doesn't. I have decided to take it out.

4. In Fig. 7B the exponent m is plotted as a function of $\langle V \rangle^{2/3}$. From all the discussion thus far I would expect it to also be a function of $\langle L \rangle$ and L_0 . Is there a physical reason why the former is more appropriate than the latter?

We believe this parameter is important as it represents the mean projected nanosheet area which intersects with the light. We have modified the text to clarify:

“This parameter, $\langle V \rangle^{2/3}$, would be expected to influence the scattering of light as it represents a characteristic area associated with the nanosheet and is a measure of the average nanosheet area presented to the incident light.”

5. As noted in line 442 it is $\langle V \rangle = \langle L \rangle \langle w \rangle \langle t \rangle / 2$. Is it possible to calculate $\langle V \rangle = \langle L w t \rangle / 2$ for the distributions? Wouldn't that be more appropriate as average volume?

The referee is of course correct. However, we have chosen to write $\langle V \rangle = \langle L \rangle \langle w \rangle \langle t \rangle / 2$ rather than the more correct $\langle V \rangle = \langle L w t \rangle / 2$ for a reason. This facilitates using the master as a metric to estimate nanosheet dimensions from m . We have added the following text to Supplementary Note 3 (SI) to clarify:

“In addition, it should be pointed out that we have written some of the dimensional parameters in a manner which is not strictly correct e.g. $\langle V \rangle = \langle L \rangle \langle w \rangle \langle t \rangle / 2$ rather than the correct version: $\langle V \rangle = \langle L w t \rangle / 2$. The reason for this is to allow the scattering exponent mastercurves to be used as size metrics. For example, as described in the main text, once m is obtained from fit, equation 7 can be used to estimate the mean nanosheet volume $\langle V \rangle$. These data can be combined to estimate the mean nanosheet thickness, $\langle t \rangle$, using $\langle V \rangle \approx \langle L \rangle \langle w \rangle \langle t \rangle / 2$. Most liquid exfoliated nanosheets have $\langle W \rangle \sim \langle L \rangle / 1.5$ (figure 3G). This allows us to estimate mean nanosheet thickness using $\langle t \rangle \sim 3 \langle V \rangle / \langle L \rangle^2$ if $\langle L \rangle$ is known (e.g. from TEM).”

6. In line 328 the correct approximation is $Q = p^2/2$ (it seems the authors expanded the sine term up to p^3 and the cosine term up to p^2 . But since the sine term is divided by p and the cosine term is divided by p^2 , they should expand the cosine term up to p^4)

The referee is absolutely correct. We have fixed this very silly error.

7. In line 381 the Rayleigh and vdH prefactors are inter-switched.

This has been fixed

8. In line 477, what is meant by “trial and error”?

Basically it means trying different functions. To avoid confusion, we have changed the text to

“We found the time dependence of the aggregate volume to be consistent with an empirical stretched-exponential-like function”

9. In line 300: it should be figure 6A

Fixed

10. In line 436 you mean Eq. 5 and 6?

Yes, this has been fixed

Reviewer #2 (Remarks to the Author):

Optical extinction spectroscopy is one important technique to characterize the properties of new molecules or nano-materials, and the extinction spectra contain contributions from both absorption and scattering. Scattering can dominate optical extinction spectra and cannot be ignored in some molecular solutions and suspensions of large nanoparticles in the range 50-1000 nm. Scattering can completely obscure absorption features hampering quantitative spectral analysis. However, many researchers are confused on extinction spectra and absorption spectra. As a result, it is quite necessary and important to distinguish these two spectra.

In this manuscript, Andrew Harvey et al. selected six different 2D materials with large bandgaps, produced each kind of nanosheets with five different sizes, and investigated the non-resonant light scattering in order to study the nanosheet parameters control the scattering coefficient spectra. They also afforded a simple model to estimate the magnitude of the scattering coefficient as a function of wavelength for different nano-materials of different sizes. They found that the scattering coefficient spectra lied roughly between the predictions of Rayleigh and van der Hulst, and with the nanosheet size increasing, the scattering power-law coefficient transitioned from Rayleigh-like to van der Hulst-like. This work affords a systematical instruction on non-resonant light scattering in nanosheet suspensions.

The work is of high novelty and well organized. I think the manuscript can be accepted for publication if the below comments can be addressed.

1. Line 100, please give the typical α value of molecular system, or afford references.

I have added a reference which give typical values

2. Please add some discussion on what relationship between the particle dimension D and the wavelength, the scattering cannot be ignored? Is it same or similar in different kind of materials?

I have added a new section entitled “Onset of scattering” which deals with this topic. Briefly, we find an equation for nanosheet length when scattering starts to become important. As an example, we show that for GaS, this yields a length of 30 nm which is sensible.

3. Does scattering change with the light intensity? Please add some comments on this. And please afford the light intensity when you measure the optical extinction, absorption and scattering spectra.

I think we may have been unclear here. All measurements were performed in a standard optical spectrometer which means at v low power. No intensity dependence is expected. To clarify, I have modified the text:

“Here, we used a standard optical spectrometer couples with integrating sphere (see SI) to measure the....”

4. In the “Applying Rayleigh and van der Hulst approximations to the non-resonant scattering regime” part, you use an approximation of $n/n_0 \sim 1$ and get equation (2b). However, figure 6A is plotted using $n=1.76$ and $n_0=1.33$. The value of n is much larger than n_0 . Please add some comments on this. If you do not use the approximation, how about the result will be?

In brief, this approximation is >80% accurate so long as $n/n_0 < 1.5$ which is always the case here. To clarify, I changed the text slightly:

“...we will use an approximation quoted by van de Hulst¹⁴ (see SI) which is reasonably accurate for $1 \leq n/n_0 \leq 1.5$, as is the case here....”

I also added a section to the SI (“Scattering parameter conversion and the approximate use of $(n/n_0 - 1)^2$ ”) which explains this in more detail and directs the reader to the literature.

5. Line 300, I am afraid the authors made a mistake in the sentence “versus λ in figure 4A”. It should be figure 6A.

Corrected

6. From figure 6B, we can see neither the two theoretical curves agree well with the experimental data. Could you please give some explanations?

As we show later in the paper, this is because the scattering observed here is midway between Rayleigh and vdH approximations. However, at this point in the paper, this may be confusing. To address this, we have added the following sentence to the paper just after where fig 6B has been described: “As we will show below, these effects are due to the fact that light scattering by liquid exfoliated nanosheets tends to fall between the limits set by Rayleigh and vdH scattering.”

7. Line 339, the authors claimed that good fits were obtained. Would you please afford several fitting results?

We have added a figure in the SI showing example fits. It is reproduced below.

Reviewer #3 (Remarks to the Author):

The article by Harvey et al proposes the use of non-resonant light scattering to estimate flake sizes of exfoliated 2D materials.

The authors have used large band gap 2D material flakes in water-surfactant suspension, with the exception of GaS where they used NMP.

The authors show that the scattering coefficient spectra of all the studied materials lie between Rayleigh and van der Hulst models, transition from the former to the latter as the nanosheet size increased following a mastercurve of power law coefficient vs nanosheet volume.

Although the article is rather long with 8 composite figures, it is written very well considering the complexity of the experimental analysis presented. Figures are also clear and legible.

However, I am not entirely convinced with the model and analysis presented and am unable to recommend the publication of the article in Nat Comms.

Below I have listed (in no particular order) some of my concerns that the authors may consider for future submissions:

* The authors consider scattering coefficient at 1 μm . Why exactly?

1 μm is an arbitrary value. If we were being strict about using SI units, the scattering coefficient would be fitted versus wavelength expressed in meters and then K would reflect the scattering coefficient at $\lambda=1\text{m}$. However, this number would have no physical relevance. Better to choose $\lambda_0=1\mu\text{m}$ and obtain a value of K representing scattering at $1\mu\text{m}$ which is a round number within the non-resonant regime. We have slightly modified the text to clarify by adding his sentence:

“(N.B. we choose the arbitrary value of 1 μm to illustrate how the scattering coefficient varies with nanosheet size at fixed wavelength in the non-resonant regime).”

Also, does 1 μm not fall within the non resonant regimes for some of the commonly studied semiconducting 2D materials too? If yes, couldn't they also be considered in this study (excluding the part of the absorption spectra associated with material bandgap)? I also note that most of the measurements are carried out till 800 nm, probably due to absorption from solvents used. This requires a clear explanation.

This is an excellent point. Firstly, 800 nm was chosen because for these materials under study, a sufficiently large wavelength window could be analysed (typically 500-800 nm). Beyond 800 nm, we sometimes struggle from experimental setup problem, as a detector change over occurs at ~ 820 nm. This sometimes gives artefacts in the spectra (such as jumps at the changeover). To avoid complications, we have therefore only analysed the spectra up to 800 nm. The solvent (water) would in principle allow acquisition up to 1380 nm.

To clarify, we have added the following text to methods in the Supporting Methods:

“N.B. spectra were measured only up to 800 nm even though the spectrometer is capable of reaching 1400 nm. This is because a lamp-change just beyond 800 nm tends to introduce artefacts which hamper analysis of the scattering spectra.”

Regarding other materials, indeed we have attempted to analyse the scattering background of other semiconductors such as TMD nanosheets, but found very scattered, unreliable results. We attribute this to the following reason: For the wide bandgap materials, we had to move relatively far from the resonant region. For example, GaS has a relatively weak absorbance at $10^3 \text{ Lg}^{-1}\text{m}^{-1}$ at $\sim 350 \text{ nm}$ and yet, the power law fits were only of goof quality for wavelengths $> 500 \text{ nm}$. This is because the power-law behaviour does not begin immediately at the bandedge but evolves as resonant effects die away. This corresponds to an energy difference of $\sim 1 \text{ eV}$. Taking MoS_2 , with an A-exciton absorbance at $\sim 660 \text{ nm}$, this would mean, only scattering data at wavelengths $> 1400 \text{ nm}$ (if keeping the same 1 eV energy difference) would be reliable. This wavelength range is not accessible in most solvents. We even attempted to increase the accessible wavelength range by using deuterated water, where measurement up to 1600 nm is possible. However, due to varying content of water condensing into the sample, this did not yield reliable data. Similar problems arose when using solvents such as NMP and CHP. The reason why it is important to move far away from the resonant region is that the scattering spectra in the resonant region follow the absorbance in shape, albeit red-shifted. The higher the oscillator strength of the material and the larger the scattering, the more problematic this gets (Nat. Comm. 5, 4576 (2014)). In addition, for very small TMD nanosheets, we observe a resonant background even in the nIR which additionally complicates the analysis (potentially edge absorbance). In this manuscript, we first wanted to establish the fundamentals of the nonresonant scattering and have thus chosen these wide bandgap materials. With this in mind, we plan to analyse the scattering closer to the resonant regime for other semiconductors in the future.

* The authors state that "As a result, the presence of this scattering background can obscure or shift the positions of absorption peaks, making detailed analysis difficult "- It is unclear why a broad background could noticeably shift the position of absorption peaks.

The reason why a pronounced scattering background is problematic for the assessment of peak positions in two-fold. The first problem is a mathematical one: when a peak is superimposed on a powerlaw background decreasing with increasing wavelength, the peak will appear red-shifted as it is the sum of peak and background. While this can widely be accounted for by analysing peak positions from second derivatives, this treatment of the spectra requires smoothing which can modify peak shape and hence position when done incorrectly (Physica Status Solidi B (2017), 254, 1700443). The second problem is that, the scattering in the resonant regime follows the absorbance in shape, but red-shifted. When scattering dominates, this adds an additional peak shift to extinction spectra compared to absorbance spectra. In such a case, spectroscopic metrics for thickness which rely on absorbance peak shifts can no longer be applied quantitatively.

* The authors state:

Line 50: "Scattering contributions manifest themselves as a broad background in the resonant

regime which falls off with increasing wavelength in the non-resonant regime"

Line 79: "scattering can dominate extinction spectra, especially in the long wavelength non-resonant regime. "

I am not sure how both the above statements could be correct.

Yes, I see how these statements appear conflicting. I have modified them to try to clarify:

“Scattering contributions manifest themselves as a broad background which falls off with increasing wavelength in the non-resonant regime.”

“As a result, for nanoparticle dispersions, scattering can dominate extinction spectra, especially in the non-resonant regime where the absorption is very weak.”

* In line 103, the authors state "However, even though equation 2 only holds for ...". Do the authors mean both 2a and 2b?

Yes both, but to clarify, I have changed it to 2A as this is the equation that appears before the text in question

* The authors include quite a detailed background on how scattering spectra has/has not been considered for 2D materials, including that of graphene. What is not clearly understood is, why these would be any different from any nanoparticles.

The authors state that the mastercurves in Fig 7 works only for disk-like particles (line 431), not spherical particles (like PS beads). It is a little confusing because the nanosheets are considered to be diamond shaped (line 403) for equation (7). Is it not sufficiently close to a spherical geometry?

More specifically, how does this model work for other nanoparticles? Why, for example, this model does not work with gold nanoparticles and polystyrene nanobeads? or even carbon nanotubes?

Firstly, the diamond reference may have been confusing. I mean a 2D geometry where the planar face is roughly diamond shaped eg:

This diamond reference has been removed. Clearly, this is not close to spherical geometry.

Secondly, the paper described an empirical model which is based on both Rayleigh and Mie scattering. Deviations from both are encoded in the kappa parameter. We see significant deviations, especially for large nanosheets. These deviations are almost certainly because the planar nanosheet geometry deviates from the spherical geometry on which the basis models are made. Of course the empirical model will work for both 1D and 0D particles (PS particle data are displayed in the SI). However, the dependence of kappa and m on particle dimensions will be different to the 2D case. We have tried to clarify this by adding the following paragraph:

“Equations 5 a&b represents a semi-empirical combination of Rayleigh and van der Hulst scattering approximations. Combined with equation 6, they describe the transition from small ($m=4$) to large ($m=2$) nanosheet behaviour. We note that for very small flakes κ is quite close to the Rayleigh prediction (figure 7a), probably because the shape of very small objects should not affect scattering dramatically. However, for large nanosheets, measured values of κ deviate significantly from the vdH approximation. We believe this a manifestation of the differences in scattering by large disc-like particles compared to that from large spheres on which the vdH model is based.”

* It is not clear whether the authors have considered the effect of absorption from the surfactants and micelles and the scattering from surfactant micelles (apart from GaS)

The surfactant used was sodium cholate which has an absorbance onset at ~ 220 nm since it is not aromatic. Absorbance spectra are thus not affected in the spectral region where the analysis was performed. In terms of scattering, the scattering strength of small surfactant micelles is significantly lower than that of colloidal nanosheet dispersions which typically show a pronounced Tyndall effect. Hence there is no observed difference between surfactant and solvent based dispersions. This can be illustrated by a graph of the scattering exponent of $\text{Ni}(\text{OH})_2$ as function of nanosheet size in water surfactant and IPA, respectively which is added to the SI in the revised version.

We have added the following text to the main manuscript: “). We note that neither the surfactant solution nor solvent displayed any observable absorption or scattering in the spectral region of interest.”

* It is not clear why GaS is the only material to be exfoliated in NMP.

GaS oxidises when exposed to water thus NMP was chosen as the media for exfoliation (Chem. Mater., 2015, 27 (9), pp 3483–3493). We have modified the text:

“The exception is GaS which was exfoliated in N-methyl-2-pyrrolidone (NMP) using a sonic bath due to its propensity to oxidise in water.”

* All of the dispersions appear cloudy in Fig 1B. If this is indeed the case, it indicates very strong scattering in the visible. Are these the XL or XS dispersions?

All are XL. We have added the following text to the caption:

“In each case, these are very large, size selected nanosheets which show strong scattering.”

* The authors state that "Such spectra are almost always erroneously presented in papers on 2D materials as “absorption spectra”.³⁶⁻³⁸ " While the statement is fully correct, it might be prudent to point out that most of the research groups working on 1 and 2D materials have made this error over the past few years and only some research groups have started to correct this as they have access to spectrometers with integrating spheres. For example, doesn't "doi:10.1038/nano.2008.215", one of the highest cited articles in liquid phase exfoliation from the same group also made this error? In my humble opinion, it is unfair to associate papers from some specific groups about this when almost everyone in the field made the same error in judgement in recent years.

This is a fair point. I have changed the citation to only cite one of our papers (and no one else).

* The authors state that "in the non-resonant regime where $\alpha \sim 0$ " (Line 280). Is this strictly true? In addition to a certain carrier concentration at the temperature where the experiment is carried out, what about edge effects? I note that edge related states in 2D nanosheets for linear optical absorption has been considered by several groups in recent years.

The reviewer is correct. However, I don't want to go beyond the scope. I have changed the text:

“...where α is generally very small....”

* The number of data points in Fig E,F,G,H seem inconsistent (e.g. 6 in E, G and 5 points in F, H) unless some data points are directly below other entries. Should't there be same number of data points as we are looking at the same dispersions and using the same AFM scans for height, width and thickness measurements?

I have checked the worksheet and all data points are present. If any appear missing it's because they are eclipsed.

* The data in Fig 7B could be fitted significantly better with a straight line. The current fit and the associated text explaining the transition from Rayleigh to vdH is not convincing. It is also unclear why the nanosheets are modelled as diamond-shaped (line 403).

A straight-line fit makes no sense physically as very small nanosheets cannot display $m > 4$ while very large nanosheet cannot display $m < 0$ as would be implied by a straight line fit. Physically, the fit must be sigmoidal with a small size limit of $m = 4$ and well defined large-size limit. The vdH model, coupled by the specific disk solution to Rayleigh Gans theory imply that this large-size limit is $m = 2$. Our data is consistent with that. We have rewritten

this section to make it more convincing. In addition the additions in response to the other referees should have made this clearer. We have removed the diamond reference as the referees' comments have made it clear that this was very confusing.

* The authors state that "For typical nanosheets with $L / W \sim 1.5$ and $L / t \sim 30$ (figure 3), V_0 would represent nanosheets with $L \sim 260$ nm, $W \sim 175$ nm and $t \sim 9$ nm ". How accurate is this estimation from the measurements performed? An indication would be useful.

Actually, we have removed this section of text to shorten the ms.

* Could the authors please explain the statement "We found all κ vs. mean nanosheet length, L , data to fall on a materials-independent master curve, characterised by $\kappa \propto L^{-2}$, rather than remaining constant as expected for spheres." Are the authors implying that κ would remain unchanged regardless of the size of spherical scatterers?

Yes, this is what the vdH model (eq 4) implies. We have modified the text to clarify:

"...rather than remaining constant as predicted by the vdH model, indicating that platelets scatter somewhat differently to spheres."

* Is there any information available on the accuracy of the spectrometer for the measured concentration at the measured wavelength range?

According to the manufacturer's website the accuracy is +/- 0.0003A up to 1A. All our measurements are below this value at the point where the scattering exponent is measured.

Others:

* Typos:

Line 165: "[ref5]"

Fixed

Line 229: "aspect ratios"

Fixed

* It would be useful to include a scale bar for Fig 2.

It is hard to fit scale bars in these multi panel figs. Instead, we have shown examples in the SI

REVIEWERS' COMMENTS:

Reviewer #1 (Remarks to the Author):

I am satisfied from the authors' changes, the article is recommended for publication in Nature Communications as it is.

Reviewer #2 (Remarks to the Author):

The authors have addressed all my questions well. I think the manuscript can be accepted for publication.

Reviewer #3 (Remarks to the Author):

All my comments have been addressed fairly.
I recommend publication of the article.